

# First fully-diurnal fog and low cloud satellite detection reveals life cycle in the Namib

Hendrik Andersen[1,2] and Jan Cermak[1,2]

[1]Karlsruhe Institute of Technology (KIT), Institute of Meteorology and Climate Research
[2]Karlsruhe Institute of Technology (KIT), Institute of Photogrammetry and Remote Sensing

**Correspondence:** Hendrik Andersen (hendrik.andersen@kit.edu)

**Abstract.** Fog and low clouds (FLC) are a typical feature along the southwestern African coast, especially in the central Namib, where fog constitutes a valuable resource of water for many ecosystems. In this study, a novel algorithm to detect FLC over land from geostationary satellite data using only infrared observations is presented. The algorithm is the first of its kind as it is stationary in time and thus able to reveal a detailed view into the diurnal and spatial patterns of FLC in the Namib region.

A validation against net radiation measurements from a station network in the central Namib reveals a high overall accuracy with a probability of detection of 94 %, a false alarm rate of 12 % and an overall correctness of classification of 97 %. The average timing and persistence of FLC seem to depend on the distance to the coast, suggesting that the region is dominated by advection-driven FLC. While the algorithm is applied to study Namib-region fog and low clouds, it is designed to be transferable to other regions and can be used to retrieve long-term data sets.

*Copyright statement.*

## 1   Introduction

Fog is commonly perceived as a hazardous weather situation that can impact traffic systems as well as the economy (Cermak and Knutti, 2009; Egli et al., 2018). In arid environments like the Namib desert, fog can act as a critical source of water that enables life for diverse species and helps sustain ecosystems (e.g., Seely, 1979; Shanyengana, 2002; Ebner et al., 2011; Azúa-

Bustos et al., 2011; Roth-Nebelsick et al., 2012; Eckardt et al., 2013; McHugh et al., 2015). As such, knowledge on the exact occurrence and spatiotemporal patterns of fog holds potential ranging from socio-economic benefits to a better understanding of fog processes and fog-driven ecosystems.

As previous studies (e.g., Cermak and Knutti, 2009; Lee et al., 2011; Egli et al., 2016; Nilo et al., 2018) have shown, geostationary satellites have the potential to draw a spatiotemporally coherent picture of the occurrence of fog and low clouds

(FLC). However, information on FLC from satellites is typically inferred using separate daytime (e.g., Bendix et al., 2006; Cermak and Bendix, 2008, 2011; Nilo et al., 2018) and night-time (e.g., Ellrod, 1995; Cermak and Bendix, 2007) algorithms, disrupting our view on fog development at a critical time of its life cycle, as typically, shortwave radiative heating starts the



dissipation of fog shortly after sunrise (Tardif and Rasmussen, 2007; Haeffelin et al., 2010; Wærsted et al., 2017). This break in retrieval techniques has thus limited the applicability of satellite-based FLC observations for the analysis of entire fog life cycles. Lee et al. (2011) have developed an approach to continuously monitor fog from a geostationary satellite platform, however, their algorithm essentially consisted of three different day, dusk/dawn and night modes and cannot be described as stationary in time. The overarching goal of this study is thus to develop and validate a single, diurnally stable satellite retrieval of FLC over land to enable the exploration of currently untapped potentials of satellite-based analyses of FLC.

Past satellite retrieval algorithms of FLC have typically consisted of a sequential application of a number of spectral tests on the basis of individual pixels in the initial classification and a subsequent merging of FLC pixels to entities (e.g., Cermak and Bendix, 2008; Egli et al., 2016). Using only spectral information, FLC can be detected using a combination of brightness temperatures in the middle infrared (MIR) and thermal infrared (TIR) (Cermak and Bendix, 2007, 2008). As during daytime, the MIR has a solar component, separate daytime and night-time retrievals are needed. A stable and fully-diurnal retrieval of FLC would thus have to solely rely on observations in the TIR. However, as Guls and Bendix (1996) state, a purely TIR-based detection of FLC is not possible, as the brightness temperatures of FLC and land surfaces are too similar.

In the realm of image analysis and machine vision, the spatial context of a pixel is often exploited for its classification (Toussaint, 1978). In recent times, the utilization of contextual information has become more frequent in environmental remote sensing techniques (e.g., Zhu and Woodcock, 2012; Bian et al., 2016; Kim et al., 2018). For FLC, distinct spatial patterns that vary over time are to be expected due to the inversion-limited homogeneous cloud top and the coherent movement of FLC patches over the Earth surface. The guiding hypothesis of this study is thus:

By a combination of spectral tests and contextual information, FLC can be robustly detected using only satellite observations in the thermal infrared and enable new insights into the diurnal patterns of FLC and its spatial variability.

In this study, FLC is detected along the south-west African coast with a specific focus on the central Namib desert, where fog is an important part of local ecosystems (Seely, 1979; Shanyengana, 2002). Knowledge on the spatiotemporal occurrence of Namib-region fog is incomplete (Cermak, 2012), and while it is commonly tied to the quasi-persistent stratiform clouds in the Southeast Atlantic (Lancaster et al., 1984; Henschel and Seely, 2008), the processes that lead to the formation of fog are still controversly discussed, as recently, Kaseke et al. (2017b) found indications for frequent water from freshwater sources in fog and related this to radiatively-driven fog formation.

## 2 Data and approach

### 2.1 Geostationary satellite observations

The main data basis for this study are observations from the most recent Spinning-Enhanced Visible and Infrared Imager (SEVIRI) onboard the Meteosat Second Generation (MSG, in this case Meteosat 11) satellite platform. While the SEVIRI instrument's measurements cover a spectral range from $0.6\mu m$ to $13.4\mu m$ with 11 channels (plus a high-resolution visible channel), here, only the calibrated brightness temperatures from four channels in the TIR are used (8.7, 10.8, 12.0 and $13.4\mu m$). The SEVIRI instrument features a repeat rate of 15 minutes (96 hemispheric scans per day), at a spatial resolution of 3 km at





nadir (Schmetz et al., 2002). The data used in this study cover the period from 2015–2017 in the region from 13.5°S–35°S and from the coast inland to 20°E.

## 2.2 Algorithm design

The FLC-detection algorithm has two parts: 1) an initial classification and 2) a contextual plausibility control of detected FLC
pixels. The initial classification of a given scene is designed as a decision tree, with sequential application of simple spectral thresholds as shown in table 1 and, if needed, the application of structural image analyses. Sequential testing stops once a class is determined, the following tests are not carried out. In the case of surface or high cloud classes, no contextual plausibility control is needed.

**Table 1.** The thresholds used for different channels and channel combinations in this study with the outcome of the initial classification. Threshold values were determined by systematic visual analysis of SEVIRI scenes and values found in literature (Cermak, 2012).

| Channel/combination | Threshold | Determined class |
|---|---|---|
| $12.0\mu m$ - $8.7\mu m$ | $< 0.5$ | High cloud |
| $12.0\mu m$ - $8.7\mu m$ | $< 1.0$ | Surface |
| $12.0\mu m$ - $8.7\mu m$ | $> 3.5$ | Surface |
| $10.8\mu m$ | $< 276$ | High cloud |
| $10.8\mu m$ | $> 293$ | Surface |
| $13.4\mu m$ - $8.7\mu m$ | $< -19$ | Surface |
| $13.4\mu m$ - $8.7\mu m$ | $> -11$ | High cloud |

The discrimination of low-level liquid-water and higher-level ice clouds with satellite observations is comparatively easy,
especially in the Namib region (Olivier, 1995). In this study, high clouds are identified based on the brightness temperature at $10.8\mu m$ as a proxy for cloud-top temperature, and the difference in $8.7\mu m$ and $12.0\mu m$ or $13.4\mu m$ brightness temperatures as an indication of ice clouds (Strabala et al., 1994). To avoid subpixel effects of higher-level cloud edges that may lead to false FLC retrievals, the surrounding pixels of detected high clouds are also classified as difficult.

After the identification of high clouds, low clouds and land surfaces need to be discriminated. As already indicated in the
introduction, the separation of FLC and land surfaces is much more difficult in the thermal infrared than in other wavelength regions (Guls and Bendix, 1996). While land surfaces can be distinguished from FLC with the described spectral tests in some regions, frequently this is not the case (cf. Fig. 1). In order to robustly separate land surfaces from FLC, additional contextual information is needed. In this case, the brightness temperature difference between at $12.0\mu m$ and $8.7\mu m$ of each scene is compared to two composites. The composites are constructed on the basis of long-term $12.0\mu m$ - $8.7\mu m$ observations and are
intended to represent land-surface structures in cloud-free conditions. The underlying assumption for the construction of the





composites is that clouds typically have lower values in the channel difference of $12.0\mu m$ - $8.7\mu m$ than land surfaces (Cermak, 2012). The two composites are built as follows:

1. A monthly composite is created by using the monthly maximum $12.0\mu m$ - $8.7\mu m$ value at each SEVIRI time slot. Of these 96 monthly time-slot maxima, the median value is used at each pixel. Monthly composites comprise seasonal variations in land surface properties, but may in some cases be prone to cloud contamination in cloudy months at cloudy locations.

2. Thus, an annual composite is constructed by taking the median of all monthly composites for each pixel. This way, potential local, seasonally occurring cloud contaminations within the monthly composites can be eliminated.

For the separation of land surfaces and FLC, each scene (Fig. 1 a)) is compared to these composites (e.g., Fig. 1 b)) by computing a structural similarity index (SSIM (Wang et al., 2004), python implementation of scikit-image (van der Walt et al., 2014)) between the scene and the composites. The SSIM consists of comparisons of luminance (averages), contrast (standard deviations of zero-centered anomalies) and structure (correlation of normalized values) of two images. In the context of this work, the SSIM compares a moving window of 5x5-pixel image sections of a given scene with the corresponding image sections of each of the composites (Fig. 1 c)). The moving window is optimized to be as small as possible and still be useful for comparing local structures. A high SSIM (in this case $> 0.4$) gives a clear indication for the pixel to be clear as illustrated in Fig. 1 c). The comparison of Fig. 1 d) and 1 e) shows the effect of introducing the SSIM test in the algorithm. It should be noted that the applicability of the SSIM is only given if a) the composite is indeed cloud free and b) the composite exhibits sufficient spatial heterogeneity in the 5x5 pixel window. Consequently, this technique is only applicable over land and not ocean surfaces (no clear, stable spatial structures in the TIR). To ensure this, two quality flags are empirically derived from the composites: Pixels are flagged when a) the coefficient of variation of the 96 monthly time-slot maxima exceeds 0.3, indicating cloud contamination, and b) when the standard deviation within a 5x5 pixel window of the monthly composite is lower than 0.1, indicating insufficient spatial heterogeneity in the specific window of the composite.

After this initial classification, a contextual plausibility control of the detected FLC pixels is conducted. The plausibility of an accurate FLC detection is estimated by analyzing neighboring pixels. Pixel classifications are changed to 'difficult', if at least five of their eight directly neighboring pixels are classified as either high cloud or surface due to the SSIM test. As changing a pixel classification to the class 'difficult' affects the result of the plausibility control in the direct pixel neighborhood, the plausibility control is iteratively repeated for all FLC pixels until no further changes are possible. After the first iteration, the control mechanism is changed slightly, so that pixels are classified as 'difficult' if more than six (instead of five) directly neighboring pixels are classified as either high cloud, surface due to the SSIM test or 'difficult' to account for potential control-inherent classification changes.

The introduction of the SSIM test leads to a relative independence of the algorithm from strict thresholds for the separation of FLC and land surfaces. As the monthly and annual composites used for the SSIM test are directly derived from the satellite observations, the algorithm is self-adjusting to the specific characteristics of the data, likely leading to a weaker sensitivity to sensor degradation or platform changes. The technique might thus hold potential for the observation of climatic changes in FLC



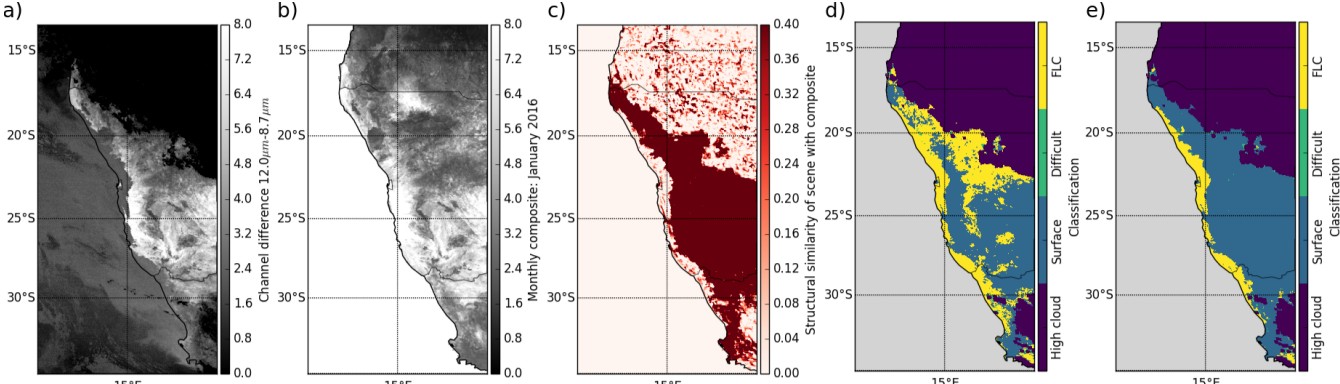

**Figure 1.** a) An exemplary scene (January 13th, 2016, 05:00 UTC) in the channel combination $12.0\mu m$ - $8.7\mu m$ is compared to b) the monthly composite of January 2016 resulting in c) the structural similarity of the scene with the composite. d) Illustration of the classification results relying on spectral tests only, and e) including the SSIM test. Quality flags are not applied in this example.

occurrence. The conceptual design of the algorithm is thought to be applicable to other regions, as long as a valid composite can be constructed from satellite infrared observations.

## 2.3 Validation approach

For the validation of the satellite-derived FLC product, three years (2015–2017) of net radiation measurements from FogNet stations in the central Namib are used. The FogNet station network comprises 11 automated meteorological stations that are aligned in two transects (N-S from 22.97°S–23.92°S and W-E from 14.46°E–15.31°E) as illustrated in Fig. 2. The stations were installed as part of the Southern African Science Service Centre for Climate Change and Adaptive Land Management (SASSCAL) initiative and offer valuable meteorological measurements in this remote region (Kaseke et al., 2017a). Of the 11 stations, 9 conduct net radiation measurements every minute using the Kipp & Zonen NR-Lite net radiometer. Station measurements are averaged between the start of two SEVIRI time slots to fit its 15-minute temporal resolution. The temporal averaging is also intended to mediate the effects of the different spatial resolutions similar to the approach in Andersen et al. (2017), as borders of advective FLC may be better captured.

For validation purposes night-time (solar zenith angle > 95°) measurements of net radiation are used to infer the presence of low-level clouds at the stations. At night, upwelling thermal radiation typically far exceeds downwelling radiation in clear conditions, whereas fog or low clouds increase downwelling radiation, leading to a nearly balanced net radiation at ground level (only situations with negative net radiation measurements are used). Figure **??** illustrates an exemplary 5-day time series of net radiation measurements at Vogelfederberg (VF) and the retrieved occurrence of FLC from co-located satellite observations. At night, the time of FLC occurrence coincides with an abrupt change in net radiation at ground level, from the range of -100 $\text{Wm}^{-2}$/-60 $\text{Wm}^{-2}$ to nearly being balanced out at 0 $\text{Wm}^{-2}$. As such, the distribution of night-time net radiation measurements is bound to be bimodal, also found in measurements (cf. Fig. 3 a)). Due to the bimodal nature of the net radiation





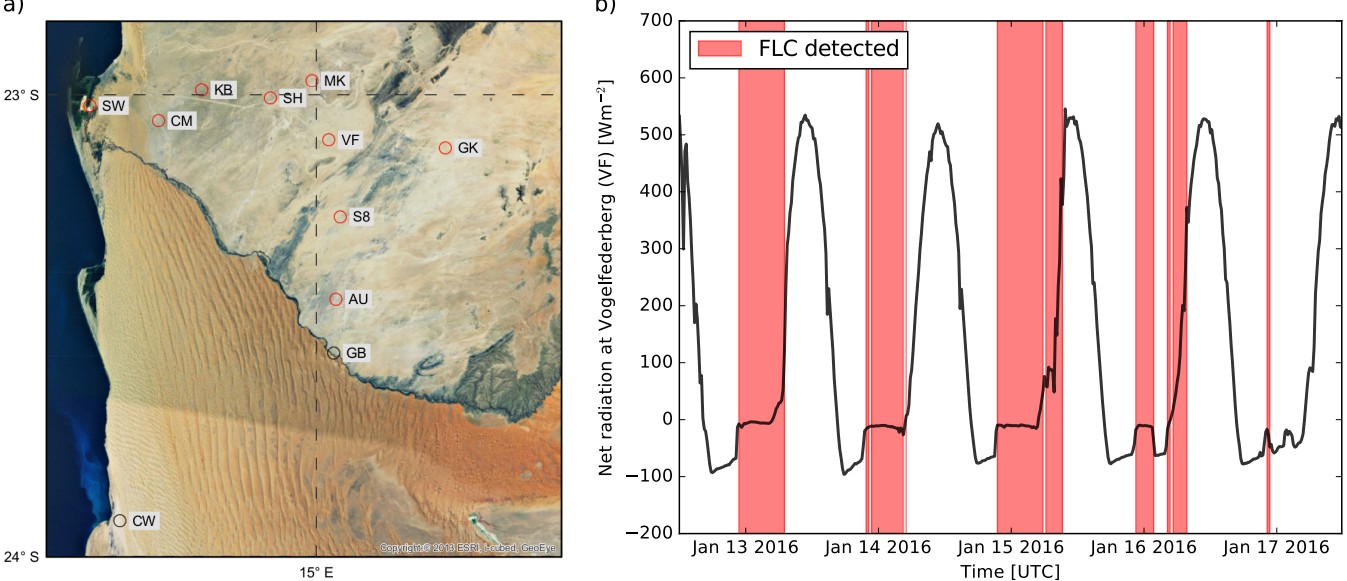

**Figure 2.** a) Locations of the FogNet stations, abbreviations (station names) are explained in the appendix. Red markers indicate the stations used for validation in this study. b) Exemplary time series of net radiation measurements at Vogelfederberg (VF). Highlighted in red are situations where the satellite algorithm has detected FLC.

measurements, a threshold can be defined at the local minimum of its smoothed histogram ((Prewitt and Mendelsohn, 1966; Glasbey, 1993), python implementation of scikit-image (van der Walt et al., 2014)) of the aggregated station measurements to separate clear and FLC situations and create a "ground truth" data set. It should be noted that the distributions of clear and FLC situations are not completely separated, and that as such the validation cannot be expected to be perfect.

5    The evaluation of the satellite-derived FLC product was performed using a set of confusion matrix tests, as is typically done (e.g., Cermak, 2012; Egli et al., 2016). By comparing the binary (FLC yes or no) information from the satellite product with the ground truth in a 2 by 2 contingency table, each satellite observation can be identified as either a "hit", "false alarm", "miss" or "correct negative". The sum of the four equals the sample size used for the validation. The following statistical measures are computed to evaluate the FLC product: probability of detection (POD - fraction of ground truth FLC that is correctly detected),

10   percent correct (PC - fraction of overall correct classifications), false alarm rate (FAR - fraction of detected FLC that are false alarms), bias score (BS - measure of bias in the classification, overestimation: BS > 1, underestimation: BS < 1), critical success index (CSI - overall measure of the correctness), and the Heidke skill score (HSS - fractional improvement of the classification over a random classification). The equations for the statistical evaluation measures are given in the appendix.



## 3 Validation of the algorithm

Figure 3 a) summarizes validation results qualitatively and quantitatively. The red histogram line shows the bimodal distribution of the aggregated night-time net radiation measurements at the nine FogNet stations for the period of 2015–2017 (excluding high cloud and 'difficult' observations), whereas the blue line represents the same station measurements, filtered for satellite-

detected FLC situations. High clouds and 'difficult' situations (grey line) are excluded from the validation as they cannot be clearly separated from neither clear nor FLC situations with surface net radiation measurements. It is apparent that most of the ground-truth FLC situations are captured by the satellite product (POD of 0.94), with only few false alarms (FAR of 0.12). The product features a high accuracy (PC of 0.97) with only a marginal positive bias (BS of 1.01). This leads to an overall high quality of the classification as expressed by a CSI of 0.83 and a HSS of 0.89. All in all, the validation is based on

325,836 co-located observations. As illustrated in Fig. 3 b), there is relatively little variation in the validation results between the different FogNet stations. The inlandmost station Garnet Koppie (GK) accounts for all outliers in Fig. 3 b) and has the highest false alarm rate. FLC occurrence frequency is thus overestimated, leading to lower overall skill. This can be attributed to the rarity of FLC occurrence in this region (cf. Fig. 4 a)). The overall accuracy of the product is considerably higher than current state-of-the-art algorithms for Europe (e.g., Cermak and Knutti, 2009; Egli et al., 2016; Nilo et al., 2018), probably in

part due to the more complex and diverse terrain and cloud structures that have to be discriminated there, and comparable to the SEVIRI-based FLC product for this region by Cermak (2012). However, the algorithm in Cermak (2012) is tailored to two specific times of the day and includes satellite observations in the visible and near-infrared spectrum. These comparisons with validation results from other studies are of an indicative nature though, as differences could also be caused by the reference data used as ground truth, or different periods considered.

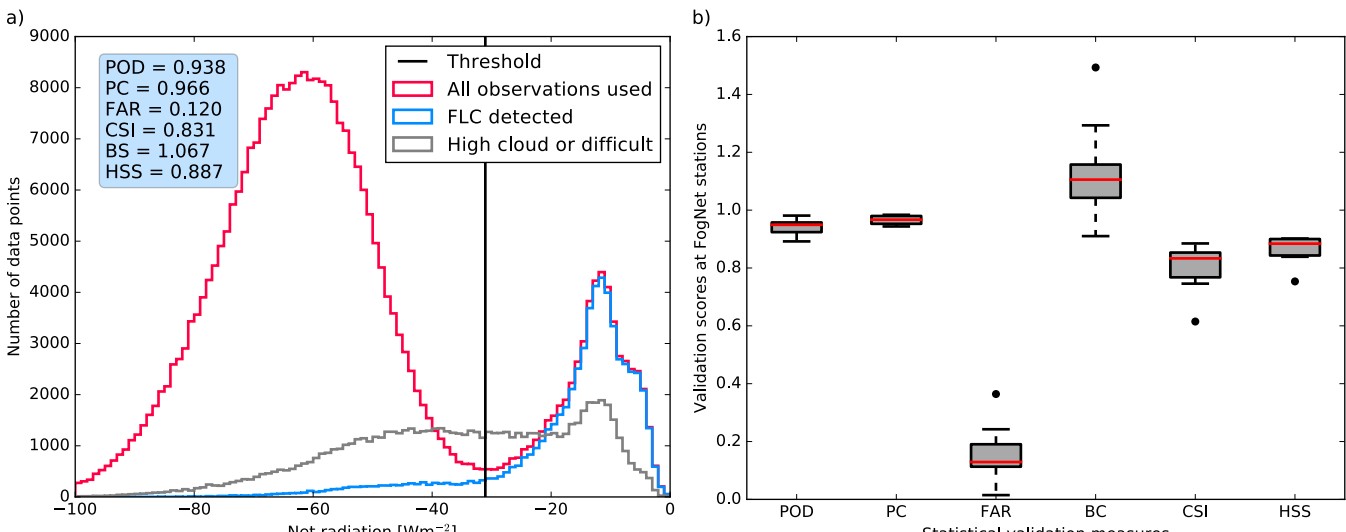

**Figure 3.** a) The aggregated validation of the satellite-derived FLC product at all stations of the time period of 2015-2017, b) the variability of validation measures across the used FogNet stations.





It should be noted that the comparison of station-level net radiation measurements with a binary satellite product is certainly not perfect. Two potential sources of error may specifically affect the validation results:

- A large difference in spatial resolution of the two observations exists. While station measurements are temporally averaged with the intent of approximating spatial variation within the area covered by a SEVIRI pixel (as in Andersen et al., 2017), the difference in field of view cannot be neglected. The difference in spatial resolution is expected to randomly lead to erroneous comparisons in both ways and thus not markedly affect the validation results. This may explain the overestimation of FLC occurrence frequency at GK, as the effect of this small random error on the validation measures scales inversly with FLC occurrence.

- Net radiation measurements are binarized in order to create a 'ground truth', even though the two modes of the distribution are not perfectly separated. For measurements close to the threshold value, an accurate discrimination of 'FLC' and 'not FLC' is not possible. This is likely to artificially impair validation results, and is manifested in a higher frequency of false alarms and misses for situations where net radiation measurements are close to the threshold value.

In addition to the validation described here, considerable effort has gone into the systematic evaluation of classifications of individual scenes as illustrated in Fig. 1. In light of these arguments, the validation results give confidence in the skill of the novel algorithm and the derived FLC product, which is well suited for the purpose of characterizing the spatial and temporal patterns of FLC in the Namib desert.

## 4 Spatiotemporal patterns of fog and low clouds in the Namib

Figure 4 a) shows the average FLC occurrence frequency in the study area over the study period. FLC most frequently occurs in the plane regions along the coastline (cf. Fig. 4 c)). Three core regions of FLC occurrence can be identified in the Angolan parts of the Namib at around 16–17°S, in a region stretching from Walvis Bay at 23°S northwards to about 18°S and at Alexander bay at around 28°S. The patterns closely resemble those found by Olivier (1995) and Cermak (2012) qualitatively and quantitatively. As illustrated by Fig. 4 b), in some regions a reliable retrieval of FLC is often not possible. This can be related to frequent occurrences of high clouds in the tropical northern parts of the study area, too little spatial variance in the clear-sky composites, as e.g. in the region of the Etendeka flood basalts at 20°S and 14°E (Bauer et al., 2000), or where the monthly composites were not stationary in time in some northeastern parts of the study area.

Figure 5 illustrates the average diurnal cycle of FLC at all FogNet stations as retrieved from satellite. It is apparent that at stations close to the coast (purple–blueish lines), FLC occurs much more frequently than further inland. The general diurnal behavior of FLC is similar at all stations, with a distinct peak of FLC occurrence at night between 3:30 UTC with 9 % relative occurrence frequency of FLC at Garnet Koppie (GK) and 6:15 UTC with 64 % at SW, and typically a fairly fast dissipation shortly after sunrise. However, distinct features of the diurnal cycle of FLC can be identified at some stations. At the station Saltworks (SW), located directly at the coastline (cf. Fig. 2), FLC tends to start to occur in the early afternoon, several hours before other stations are overcast. This could potentially be related to local land-sea winds that transport FLC from the ocean





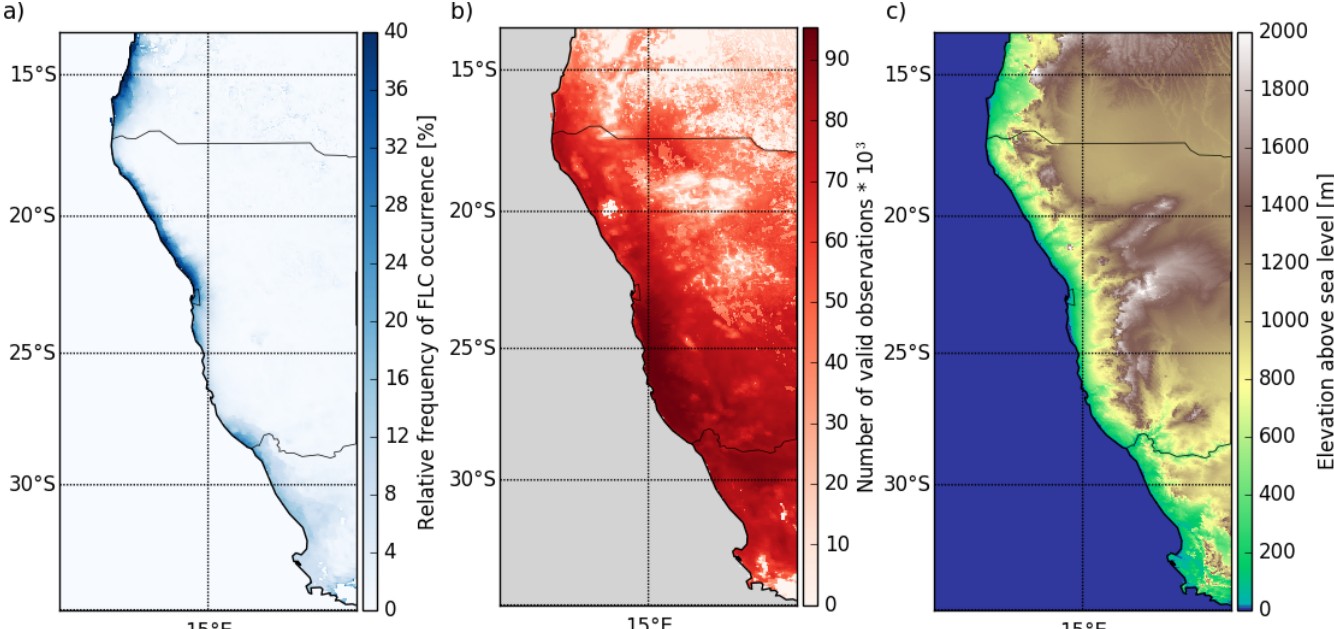

**Figure 4.** a) Average relative frequency of occurrence of FLC along the southwestern African coast during the period 2015–2017. b) Number of observations where the retrieval of FLC is possible (i.e., no high clouds and no composite-related quality flags). c) Digital elevation model of the study area.

over land during this time. In general, a time lag exists in the average start of the diurnal cycle, as well as the time of the diurnal maximum occurrence frequency that seem to be dependent on the longitudinal position of the stations, which approximates their distance to the coastline. This time lag is indicative of a region that is generally dominated by FLC that forms at the coast or over the ocean and is then advected inland, as described by Olivier and Stockton (1989), Olivier (1995) and Eckardt et al. (2013), contrasting more recent findings by Kaseke et al. (2017b). As FLCs are advected inland, locations close to the coastline are overcast first. At the coast, FLC not only occurs earlier in the day, but typically also persists longer than further inland, where FLC tends to start dissipating before sunrise. The dissipation rate (negative $\delta$ FLS occurrence/$\delta$ time in the morning hours) at inland stations is considerably lower than closer to the coast, which can probably be attributed to stronger solar irradiance later in the day. It should be noted that the figure represents highly aggregated information on the average diurnal cycle of FLC and does not preclude other formation or dissipation processes.

As a detailed statistical analysis of the full life cycle of FLC and its seasonal behavior is not within the scope of this paper, this example is intended to illustrate the potential of the novel algorithm for the analysis of spatiotemporal patterns of fog and low clouds over land.




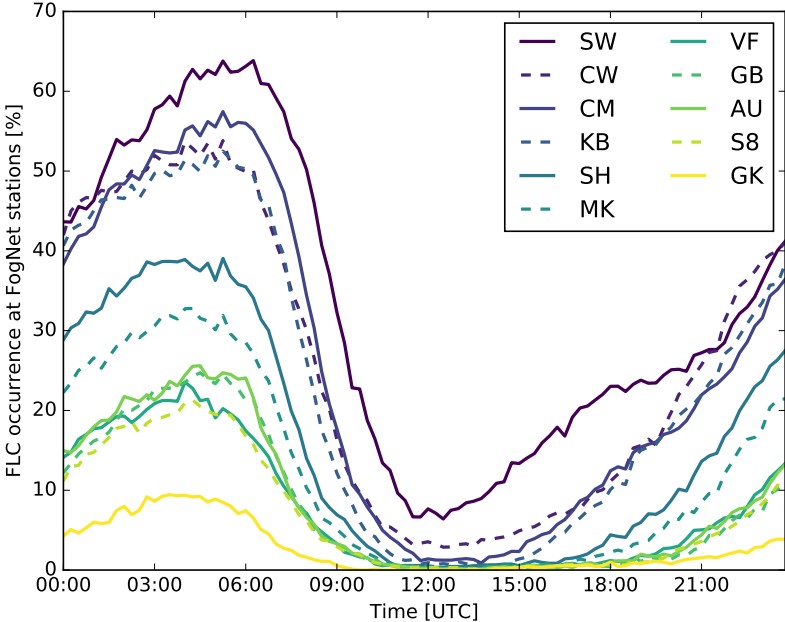

**Figure 5.** Average diurnal cycle of FLC occurrence at FogNet stations. Lines are colored by the longitudinal rank of each specific station, with coastal stations in dark colors and stations further inland in brighter colors. Every other line is dashed with the only purpose of helping their visual discrimination.

## 5    Conclusions and Outlook

The central aim of this study was to develop the first thermal-infrared-only and thereby diurnally stable satellite retrieval of fog and low clouds. The algorithm design uses a combination of spectral tests and contextual information in order to retrieve FLC. A structural similarity index (Wang et al., 2004) is computed comparing each satellite scene with cloud-free composites

to discriminate between land surfaces and FLC. The novel algorithm is thereby relatively independent from exact spectral thresholds and thus has the potential to be easily applied to other regions and to generate climate data sets. An operational deployment is possible with small adjustments in algorithm design and holds potential for the prediction of FLC dissipation. In the future, the value of the derived FLC product may be further enhanced with a retrieval of cloud-base altitudes for the separation of low-level clouds from ground fog.

The algorithm was applied to detect spatial and temporal patterns of Namib-region FLC and was validated against net radiation measurements at the FogNet station network located in the central Namib region. The algorithm shows good overall detection accuracy, with few false alarms and a small positive bias, and relatively little station-to-station variation. FLCs most frequently occur close to the southwest-African coastline, with Walvis Bay among the core regions, confirming findings from Olivier (1995) and Cermak (2012). The diurnal cycle of FLC is described for the locations of the FogNet stations. Marked

differences in the timing of FLC occurrence and temporal persistence are found. The time lag of FLC occurrence from the coast





to inland regions may be attributed to the advection of FLC from the coast inland, a typical feature of the region (Henschel and Seely, 2008; Olivier, 1995). FLC persists longer in coastal regions than further inland and but then dissipates more rapidly after sunrise.

The study shows the potential of the diurnal FLC algorithm to study fog and low-cloud patterns, processes and life cycles.

Future research efforts should focus on coherently mapping diurnal characteristics of FLC, further understanding fog formation processes specifically considering knowledge on the factors that drive low-level clouds in the Southeast Atlantic (e.g., Adebiyi et al., 2018; Andersen and Cermak, 2015; Fuchs et al., 2017, 2018), and potentially detecting changes in FLC occurrence. This may best be achieved by combining the satellite retrievals with numerical modeling and ground-based observations, and will be undertaken within the ongoing research project Namib Fog Life Cycle Analysis (NaFoLiCA).

*Code and data availability.* Code and data are available on request.

**Appendix A: Equations of statistical validation measures**

$\text{POD} = \frac{a}{a+c}$

$\text{PC} = \frac{a+d}{a+b+c+d}$

$\text{FAR} = \frac{b}{a+b}$

$\text{CSI} = \frac{a}{a+b+c}$

$\text{BS} = \frac{a+b}{a+c}$

$\text{HSS} = \frac{2(ad-bc)}{(a+c)(c+d)+(a+b)(b+d)}$

with a = number of hits, b = number of false alarms, c = number of misses and d = number of correct negatives

**Appendix B: Abbreviations of FogNet stations**

Aussinanis: AU

Coastal Met: CM

Conception Water: CW

Garnet Koppie: GK

Gobabeb Met: GB

Kleinberg: KB

Marble Koppie: MK

Saltworks: SW

Sophies Hoogte: SH

Station 8: S8

Vogelfederberg: VF

*Author contributions.* H. Andersen had the conceptual idea for the algorithm, developed the method and wrote the software, obtained and analyzed the data sets, conducted the original research and wrote the manuscript. J. Cermak contributed to method design, manuscript
5  preparation and the interpretation of findings.

*Competing interests.* The authors declare that they have no conflict of interest.

*Acknowledgements.* Funding for this study was provided by Deutsche Forschungsgemeinschaft (DFG) in the project Namib Fog Life Cycle Analysis (NaFoLiCA), CE 163/7-1. The authors would like to thank the Gobabeb Research and Training Centre for access to the station measurements and gratefully acknowledge Folke Olesen, Frank Göttsche, Mary Seely and Roland Vogt for their efforts in the field. The
10 authors also thank Julia Fuchs, Frank Göttsche, Folke Olesen, Roland Stirnberg and Roland Vogt for helpful discussions.



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
