# Peer review of "First fully-diurnal fog and low cloud satellite detection reveals life cycle in the Namib"

_Atmospheric Measurement Techniques, 2018_

## Referee Comment (RC1) · Anonymous Referee #1 · 6 Aug 2018

A good algorithm to obtain the detection of fog and low cloud using the temporal continuity of MSG-SEVIRI data, demonstrated as with high potential for use the product for many applications (for example agriculture). This work represents a novelty because it permits to detect fog and low cloud continuously during day and night with very high performances in terms of statistics. In general, I find this technical paper worth publishing. The technical part is well written, very clear, fast to read. Validation and result sections are very well deduced by the authors. Comments (given below) are suggested to be followed.

Page 2 Line 26 – From my point of view would be worth to give a brief description of the paper sections.

Page 3 Line 1 – I would rewrite the sentence in this way: " The data used in this study

cover the period 2015-2017 in the region 13.5 °s-35°S and ???-20°E."

Page 3 Line 6 – Please explain better what is intended for " . . . if needed, the application of structural image analyses." Are there cases in which this analyses can be skipped? If yes maybe can be useful to report it/them.

Page 3 Line 7 – "Contextual plausibility control" is the same operation that "structural image analyses"? Please resolve this (from my point of view) ambiguity.

Page 3 Table 1 – In Table 1, please specify threshold measure unit (I think is Kelvin degree).

Page 5 Line 16 – I think Figure ??? is Figure 2b), please correct it.

Page 8 Line 20 – in order to improve the comprehension, I think is good to draw highlight the edges of the three core regions in Figure 4.

---

## Referee Comment (RC2) · Anonymous Referee #2 · 7 Aug 2018

My review and comments will focus on four area: general overview, methodology, validation, and product impact. The paper is well written and presents an interesting approach to a challenging problem. Overview The authors present an interesting approach and methodology to create a fog and low cloud product. The application of interest stated by the authors is fog detection that is hazardous to traffic and the potential for economic impact, and the need to understand the formation and dissipation processes over the region. Does the algorithm differentiate between fog and low clouds (low clouds may not reduce visibility to the same extent as the fog)? What portion of cases can be isolated or identified as fog versus low clouds? Does the FogNet stations help to isolate and identify and differentiate fog from low clouds? The goal to develop a common algorithm that works well particularly during the transition from night to day

in order to monitor fog development and dissipation with solar insolation is admirable. The authors point to other studies that utilize different approaches during the night and day, but do not show any failure of these approaches to properly detect the life cycle of the fog. Are the authors aware of more recent work to produce a stable and fully diurnal approach for the detection of fog and low clouds with the 24 hour Red-Green-Blue (RGB) microphysics products (developed and applied to SEVIRI and GOES ABI data) using only the 8, 11, and 12 micrometer channels on these instruments? https://weather.msfc.nasa.gov/sport/training/quickGuides/rgb/QuickGuide_24hrMicroRGB_NASA_SPoRT.pdf https://www.eumetsat.int/website/home/Data/Training/TrainingLibrary/DAT_2044069.html Or NOAA's low cloud and fog product? https://www.goes-r.gov/products/opt2-low-cloud-fog.html Recognizing this work or acknowledging these other approaches should be done. Methodology This is an interesting 2 step approach which eliminates high clouds and then identifies fog and low cloud regions. The temporally varying compositing approach to represent cloud-free scenes over land as a reference is good and has been successfully demonstrated for other cloud detection approaches. The SSIM approach to identify regions that are significantly different from the cloud-free composite is interesting although limits application to ocean coastal regions where sea surface temperature structure is limit. It would be interesting to know how the threshold (0.4) and the window size were determined. The assignment of pixels as "difficult" on the edge of fog and low cloud regions in the contextual plausibility control step seems a bit subjective. While the approach is meant to address sub-pixel issues, other issues could be coming into play (marginal thermal structure in composite, complete pixel coverage if thin or dissipating fog, etc.). Eliminating these regions makes the regions identified as fog and low cloud more limited. These "difficult" pixels also seem to be eliminate from the validation section improving statistical performance of the algorithm. Additional justification is necessary for this approach. Reason for iteration of plausibility control is not clear. Can you elaborate? Validation Only night-time results are presented. A proposed strength of the algorithm is its day and night performance (?) to monitor dissipation of the fog with solar insolation. How do the day-time results

compare to these? Labeling pixels on edges of clouds as "difficult" helps the validation statics. What to the results look like if you add in results from the "difficult points" What percentage of fog pixels to difficult ones? Is there performance variability by year or by season? This would add confidence to the use of the product for climate studies. Good discussion of the potential source of errors.

Product impact on science Interesting and useful inference of spatial and diurnal variation in occurrence of FLC. Could you use a monthly varying composite to increase FLC frequency over the region?

Other things I cant locate the grey line in Figure 2a. Figure 3b it is not obvious that the dot corresponds to the values from GK. Please explain this and the error bars in the figure. The label "BC" should be BS in Figure 3b.
* * *

---

## Author Comment (AC2) · 22 Aug 2018

**First fully-diurnal fog and low cloud satellite detection reveals life cycle in the Namib**
**— RESPONSE TO REVIEWER 2 —**

Hendrik Andersen and Jan Cermak contact: hendrik.andersen@kit.edu

We would like to thank referee 2 for her/his review of the manuscript and her/his constructive criticism. Comments by the referee are colored in blue, our replies or comments are colored in black.

My review and comments will focus on four area: general overview, methodology, validation, and product impact. The paper is well written and presents an interesting approach to a challenging problem.

**Overview**

The authors present an interesting approach and methodology to create a fog and low cloud product. The application of interest stated by the authors is fog detection that is hazardous to traffic and the potential for economic impact, and the need to understand the formation and dissipation processes over the region. Does the algorithm differentiate between fog and low clouds (low clouds may not reduce visibility to the same extent as the fog)? What portion of cases can be isolated or identified as fog versus low clouds? Does the FogNet stations help to isolate and identify and differentiate fog from low clouds?

We agree with referee 2 that the differentiation of fog and low clouds is very important for both economical and ecological aspects. The algorithm presented in this work does not differentiate between fog and low clouds, hence the abbreviation FLC (fog and low clouds). As this differentiation is one of the main remaining challenges in the satellite-based remote sensing of fog, we are currently working on this using both ground- and space-based active remote sensing as well as the available FogNet station data. As of now, we cannot give a reliable estimate of the fraction of fog in the FLC product, which may vary by location, season and time of day. As such, a differentiation of fog and low clouds is beyond the scope of this work, but will be addressed in future studies.

We have included the sentence "*It should be noted that the algorithm presented here does not differentiate between ground fog and low-level clouds.*" at the end of the first paragraph of section 2.2 for clarity (in the outlook, we already mention that a retrieval of cloud-base altitudes for the separation of low-level clouds from ground fog is needed).

The goal to develop a common algorithm that works well particularly during the transition from night to day in order to monitor fog development and dissipation with solar insolation is admirable. The authors point to other studies that utilize different approaches during the night and day, but do not show any failure of these approaches to properly detect the life cycle of the fog. Are the authors aware of more recent work to produce a stable and fully diurnal approach for the detection of fog and low clouds with the 24 hour Red-Green-Blue (RGB) microphysics products (developed and applied to SEVIRI and GOES ABI data) using only the 8, 11, and 12 micrometer channels on these instruments? https://weather.msfc.nasa.gov/sport/training/quickGuides/rgb/QuickGuide\_24hrMicroRGB\_NASA\_SPoRT.pdf, https://www.eumetsat.int/website/home/Data/Training/TrainingLibrary/DAT\_2044069.html. Or NOAA's low cloud and fog product? https://www.goes-r.gov/products/opt2-low-cloud-fog.html. Recognizing this work or acknowledging these other approaches should be done.

We agree with referee 2 that the limitations of day and nighttime FLC detection algorithms could be stated more clearly. Nighttime detection of FLC has been achieved in many studies since the 1980s (e.g. Eyre et al., 1984; Bendix, 2002; Cermak and Bendix, 2007), which typically rely on the difference between a thermal ($\approx 11\ \mu$m) and mid-infrared (3.9 $\mu$m) channel. However, as Cermak and Bendix (2008) state: "*During daytime, however, the situation*

*is entirely different. The solar signal that mixes into the 3.9 μm radiation*

*renders the method useless after sunrise, as the small fog droplets reflect at this*

*wavelength. Therefore an altogether different approach is needed for daytime*

*fog detection.*"

These current day time techniques typically do not work at low solar elevation angles, which is illustrated by the following examples:

- The daytime algorithm developed by Nilo et al. (2018) works only in situations with solar zenith angle $> 85°$.

- The daytime algorithms developed by Cermak and Bendix (2008) and Cermak and Bendix (2011) work only in situations with solar zenith angle $> 80°$.

- Similarly, Guls and Bendix (1996) state that "*Unfortunately, at low sun elevations (with θ close to 90°) cos(θ) [solar zenith angle] approaches zero and the normalised grey level approximes to infinity. Therefore, normalisation is limited to sun elevations of about $10°$ (Saunders, 1985).*"

To summarize, separate day and nighttime algorithms are necessary, with neither one working sufficiently well at low solar elevation angles.

While we are aware of the qualitative products (false color composites) produced by the Eumetsat, NASA and NOAA, which are a nice tool for visualization purposes, these are not products well-suited for quantitative analyses and were thus not mentioned. As we agree with referee 2 that these sets of products might be of interest to the reader, we do mention false color imagery now with the following sentence in the introduction: "*While for visualization purposes, 24-hour false color image products may be used in case studies, these images are not well-suited for quantitative analyses.*"

**Methodology**

This is an interesting 2 step approach which eliminates high clouds and then identifies fog and low cloud regions. The temporally varying compositing approach to represent cloud-free scenes over land as a reference is good and has been successfully demonstrated for other cloud detection approaches. The SSIM approach to identify regions that are significantly different from the cloud-free composite is interesting although limits application to ocean coastal regions where sea surface temperature structure is limit. It would be interesting to know how the threshold (0.4) and the window size were determined.

The moving window is optimized to be as small as possible and still be useful for comparing local structures. The size of the moving window, as well as the threshold for the SSIM were optimized empirically, by analyzing many individual scenes. We have now mentioned this more clearly in the manuscript: "*The size of the moving window, as well as the threshold for the SSIM were optimized empirically.*"

We would argue that the approach is not limited to coastal regions (it should work in any continental region with enough spatial variance in the composites), but it will certainly not work over ocean.

The assignment of pixels as "difficult" on the edge of fog and low cloud regions in the contextual plausibility control step seems a bit subjective. While the approach is meant to address sub-pixel issues, other issues could be coming into play (marginal thermal structure in composite, complete pixel coverage if thin or dissipating fog, etc.). Eliminating these regions makes the regions identified as fog and low cloud more limited. These "difficult" pixels also seem to be eliminate from the validation section improving statistical performance of the algorithm. Additional justification is necessary for this approach. Reason

for iteration of plausibility control is not clear. Can you elaborate?

This is an interesting point for discussion. The contextual plausibility control and the class "difficult" were created during the visual quality assessment of the algorithm of single scenes. It became apparent that sometimes, at the edges of high clouds, the algorithm can misclassify pixels as FLC. This is probably related to sub-pixel cloudiness of the high clouds that can lead to a spectral signature similar to FLC/surface. The SSIM test does not find a strong similarity with the composite, as part of the region that is evaluated is overcast with high clouds. This led to the idea of the contextual plausibility control that is designed to address this issue. It specifically looks for these situations (more than half of the pixels in the immediate neighborhood are classified as high-cloud) and, if true, labels the pixel of interest as difficult. An iterative approach is chosen, as changing the class of one pixel changes the neighborhood of all its neighboring pixels, which needs to be accounted for.

We now discuss this in more detail in the manuscript: "*A situation in which this approach may fail is at higher-level cloud edges. These pixels can be have a similar spectral signature to FLC and can pass the SSIM test, as the partly overlying high cloud reduces the similarity with the composites. To avoid such misclassifications, a contextual plausibility control of the detected FLC pixels is conducted after the initial classification.*"

Validation Only night-time results are presented. A proposed strength of the algorithm is its day and night performance(?) to monitor dissipation of the fog with solar insolation. How do the day-time results compare to these?

The validation is limited to nighttime measurements, as the net radiation measurements can be binarized rather easily during night (Fig. 3a)). This is not the case during daytime, where this would have to be done for each solar zenith angle and would still be associated with higher uncertainties. We
argue that this is legitimate, as none of the channels used and no component
of the retrieval technique is physically affected by solar radiation. Thus, from
a physical point-of-view, there is no reason why the algorithm should work
differently during day time. We have looked into a large number of scenes and
found no effect of the time of day on the retrieval.

Labeling pixels on edges of clouds as "difficult" helps the validation statics.
What to the results look like if you add in results from the "difficult points"
What percentage of fog pixels to difficult ones?
Over the entire data set, the plausibility control 'corrects' about 3 % of the
detected FLC pixels and sets their class to 'difficult'. As such, it only marginally
affects the quantitative validation results as presented in Fig. 1. The right-hand
panel shows the validation where the class 'difficult' is analyzed as if it were
classified as FLC, only leading to a slightly higher false alarm rate, with the
POD and PC virtually unchanged.

[Figure]

Figure 1: The validation of the algorithm as in the submitted manuscript (left)
and computed without the use of the structural plausibility control (right).

However, thin cloud edges of higher-level clouds may lead to similar surface-
measured net radiation as FLC, making the quanitative analysis of these pixels with net radiation measurements difficult. A detailed visual analysis of a large number of individual scenes has shown an improved performance at the edges of higher-level clouds using the plausibility control.

Is there performance variability by year or by season? This would add con- fidence to the use of the product for climate studies. Good discussion of the potential source of errors.

We have computed the validation as suggested by referee 2. There does not seem to be a marked yearly variability in the performance of the algortithm as illustrated by Fig. 2.

[Figure]

Figure 2: The validation of the algorithm computed separately for the three years 2015 (left), 2016 (center) and 2017 (right).

We also computed the validation on a seasonal basis (cf. Fig. 3), with little variation of the probability of detection and percentage correct of the classification. There does seem to be a seasonal variation in the false alarm rate, which can likely be attributed to the overall occurrence frequency of FLC

as outlined in the manuscript (concerning the station GK). If only few FLC

situations occur, a (small) randomly occurring misclassification has a relatively large impact. This explains the outliers of the false alarm rate of the inland station GK, as well as the relatively high false alarm rate in the season of

March, April and May, where FLC occurs much less frequently. This is already described in the manuscript: "[...] the effect of this small random error on the

*validation measures scales inversly with FLC occurrence."*

The results underline the applicability for climate studies. We now discuss this in more detail in the manuscript.

[Figure]

Figure 3: The validation of the algorithm computed separately for the seasons December, January, February (top left), March, April, May (top right), June July, August (bottom left) and September, October, November (bottom right).

**Product impact on science**

Interesting and useful inference of spatial and diurnal variation in occurrence of FLC. Could you use a monthly varying composite to increase FLC frequency over the region?

In the current algorithm, we use two composites: a monthly and a yearly composite. We have also tested daytime-specific composites, but found no improvement in the performance of the algorithm.

**Other things**

I can't locate the grey line in Figure 2a.

Thank you for pointing this out, this referred to an old version of the figure and is now deleted from the manuscript.

Figure 3b it is not obvious that the dot corresponds to the values from GK. Please explain this and the error bars in the figure.

We have now included this information in the caption of Fig. 3b).

The label "BC" should be BS in Figure 3b.

Yes, this is now corrected in the manuscript.

**References**

Bendix, J. (2002). A satellite-based climatology of fog and low-level stratus in Germany and adjacent areas. *Atmospheric Research*, 64(1-4):3–18.

Cermak, J. and Bendix, J. (2007). Dynamical Nighttime Fog/Low Stratus Detection Based on Meteosat SEVIRI Data: A Feasibility Study. *Pure and Applied Geophysics*, 164(6-7):1179–1192.

Cermak, J. and Bendix, J. (2008). A novel approach to fog/low stratus detection using Meteosat 8 data. *Atmospheric Research*, 87(3-4):279–292.

Cermak, J. and Bendix, J. (2011). Detecting ground fog from space – a microphysics-based approach. *International Journal of Remote Sensing*, 32(12):3345–3371.

Eyre, J. R., Brownscombe, J. L., and Allam, R. J. (1984). Detection of fog at night using Advanced Very High Resolution Radiometer (AVHRR) imagery.

*Meteorological Magazine*, 113:266–271.

Guls, I. and Bendix, J. (1996). Fog detection and fog mapping using low cost

Meteosat-WEFAX transmission. *Meteorological Applications*, 3:179–187.

Nilo, S., Romano, F., Cermak, J., Cimini, D., Ricciardelli, E., Cersosimo, A.,

Di Paola, F., Gallucci, D., Gentile, S., Geraldi, E., Larosa, S., Ripepi, E., and

Viggiano, M. (2018). Fog Detection Based on Meteosat Second Generation-

Spinning Enhanced Visible and InfraRed Imager High Resolution Visible

Channel. *Remote Sensing*, 10(4):541.

---

## Author Comment (AC1)

**First fully-diurnal fog and low cloud satellite detection reveals life cycle in the Namib**
**— RESPONSE TO REVIEWER 1 —**

Hendrik Andersen and Jan Cermak

contact: hendrik.andersen@kit.edu

We would like to thank referee 1 for her/his review of the manuscript and her/his constructive criticism. Comments by the referee are colored in blue, our replies or comments are colored in black.

A good algorithm to obtain the detection of fog and low cloud using the temporal continuity of MSG-SEVIRI data, demonstrated as with high potential for use the product for many applications (for example agriculture). This work represents a novelty because it permits to detect fog and low cloud continuously during day and night with very high performances in terms of statistics. In general, I find this technical paper worth publishing. The technical part is well written, very clear, fast to read. Validation and result sections are very well deduced by the authors. Comments (given below) are suggested to be followed.

Page 2 Line 26 – From my point of view would be worth to give a brief description of the paper sections.

We agree with referee 1 that a brief structural overview can enhance the clarity of the manuscript and have now included this at the end of the introduction: "*This work is structured as follows: The data used and the novel FLC-detection technique are described in section 2, a statistical evaluation of the algorithm is given in section 3. Spatiotemporal patterns of fog and low clouds in the Namib are presented in section 4, conclusions and an outlook are given in section 5.*"

Page 3 Line 1 – I would rewrite the sentence in this way: " The data used in this study cover the period 2015-2017 in the region $13.5°$ s-$35°$ S and ???-$20°$ E."

We have adjusted the text passage to say: "*The data used in this study cover the period from 2015–2017 in the region from $13.5°S$–$35°S$ and from*

$10^{\circ}\,E$–$20^{\circ}\,E$, considering only regions over land."

Page 3 Line 6 – Please explain better what is intended for "...if needed, the application of structural image analyses." Are there cases in which this analyses can be skipped? If yes maybe can be useful to report it/them.

The structural image analyses are only needed if none of the spectral tests is true. The follow-up sentence tries to communicate this: "*Sequential testing stops once a class is determined, the following tests are not carried out.*" However, we agree with referee 1 that this may be stated more clearly in the initial sentence and have rephrased it as follows: "*The initial classification of a given scene is designed as a decision tree, with sequential application of a) simple spectral thresholds as shown in table 1 and, b) if none of the spectral tests is true, the application of a structural image test.*"

Page 3 Line 7 – "Contextual plausibility control" is the same operation that "structural image analyses"? Please resolve this (from my point of view) ambiguity.

The contextual plausibility control is one of two structural image analysis techniques used in this work. To clarify:

The initial classification is the two step technique that constists of a) spectral thresholds and b) the SSIM test. The contextual plausibility control is intended to review the results of the algorithm and check the plausibility of the detected FLC pixels. Both (SSIM test and contextual plausibility control) can be described as structural image analyses, as is done on Page 3 Line 6.

We agree with referee 1 that this can be stated more clearly and have rephrased said text passage: "*The initial classification of a given scene is designed as a decision tree, with sequential application of a) simple spectral thresholds as*

*shown in table 1 and, b) if none of the spectral tests is true, the application*

*of a structural image test. Sequential spectral testing stops once a class is*

*determined, the following tests are not carried out. The additional contextual*

*plausibility control is only tested for FLC pixels."*

Page 3 Table 1 – In Table 1, please specify threshold measure unit (I think is Kelvin degree).

Thank you for pointing this out, units are now included in the table.

Page 5 Line 16 – I think Figure ??? is Figure 2b), please correct it.

Yes, indeed. This is now corrected in the manuscript.

Page 8 Line 20 – in order to improve the comprehension, I think is good to draw highlight the edges of the three core regions in Figure 4.

While we understand the point of referee 1, and agree that highlighting the core regions would help identifying them in the figure, we would like to keep the figure as it is. Our argument is that drawing the regions directly into this results figure would, in our opinion, already introduce somewhat of an interpretation and potentially bias a reader.